# Comparing diagnostic methods for historical arbovirus outbreaks: Insights from 19th century *"dengue"* epidemics

Timothee Bonifay[1,2]*, Mathieu Nacher[1,2], Clémence Bonnefoy[1], Benjamin Rossi[3], Edouard Hallet[1], Philippe Abboud[1], Guillaume Bellaud[4], Nathalie Dournon[5], Aurélia Henn[6], Adrien Lemaignen[7], Liem Binh Luong Nguyen[8], Diane Sanderink[9], Gaelle Walter[1], Alexandre Bleibtreu[10], Loïc Epelboin[1,2]

**1** Centre Hospitalier de Cayenne, Cayenne, French Guiana, **2** INSERM UA17 Santé des Populations, Université de Guyane, Cayenne, French Guiana, **3** Centre Hospitalier Intercommunal de Robert Ballanger, Aulnay-sous-bois, France, **4** Centre Hospitalier de Montauban, Montauban, France, **5** Hôpital Avicenne, AP-HP Hôpitaux Universitaires Paris Seine-Saint-Denis, Paris, France, **6** Centre Hospitalier Sud Francilien, Corbeil Essonnes, France, **7** Centre Hospitalier Universitaire de Tours, Tours, France, **8** Hôpital Cochin, AP-HP. Centre - Université de Paris, Paris, France, **9** Service des Maladies Infectieuses et Tropicales, Centre Hospitalo universitaire d'Anger, Angers, France, **10** Hôpital de la Pitié Salpêtrière, AP-HP Sorbonne Université, Paris, France

\* timothee.bonifay@gmail.com, timothee.bonifay@ch-cayenne.fr

## Abstract

This study reexamines historical *"dengue"* epidemics of the 18th and 19th centuries, suggesting that chikungunya virus (CHIKV), rather than dengue virus (DENV), may have been responsible for many of these outbreaks. While *"dengue"* was identified as a tropical disease in the 19th century, its exact characteristics remain unclear, and some descriptions align more closely with CHIKV, known for its distinctive symptom of joint pain. Three approaches were used to investigate these historical epidemics: **(A) Expert opinion** provided a contextualized and comprehensive analysis but faced criticism for its subjective nature **(B) Clinical Score Application** allowed for greater objectivity and a more stringent diagnosis, although it was limited by the need for a complete description **(C) Double Proofreading:** Two expert groups independently reviewed the articles, which enhanced objectivity but also led to greater variability in the results. The study suggests that CHIKV was mainly responsible for the epidemics historically attributed to *"dengue"*. It highlights the challenges of diagnosing diseases from historical records. It also raises the possibility that other alphaviruses, like mayaro virus, could be involved, but CHIKV remains the primary candidate. This study offers intriguing insights into pathogen identification in historical epidemics, emphasizing the importance of a combined approach for a more precise understanding of past diseases and their evolution.

**Data availability statement:** All data used in the study are extracted from published historical sources, which are fully referenced in the bibliography of the manuscript.

**Funding:** The author(s) received no specific funding for this work.

**Competing interests:** The authors have declared that no competing interests exist.

## Author summary

In the 18th and 19th centuries, several large epidemics were described under the name "dengue," but their true cause remains uncertain. At that time, neither the dengue virus (DENV) nor the chikungunya virus (CHIKV) had been discovered. Yet the symptoms reported in historical records—especially severe joint pain—often resemble what is now known to be caused by CHIKV. In this study, we reassessed the cause of these historical outbreaks using three complementary methods: expert contextual review, a standardized clinical scoring system, and independent double-review by two teams of physicians. Each method has strengths and limitations, and while results varied somewhat, all pointed toward CHIKV as the most likely cause of many of these past epidemics. We also considered whether other viruses, such as Mayaro virus, might have been involved, but found little evidence to support this. Our study highlights the challenges of diagnosing diseases in the absence of laboratory tests and underlines how modern clinical knowledge can help reinterpret historical data. This approach offers new insights into the origins and spread of arboviral diseases and helps improve our understanding of how similar epidemics have occurred and evolved over the past two centuries.

## Introduction

The 19th century was a pivotal period for medical discoveries. The germ theory of diseases gained widespread acceptance, epidemiology emerged as a scientific discipline, and the medical lexicon was standardized through numerous treatises and dictionaries. In the context of colonization and globalization, infectious and tropical diseases such as cholera, plague, and yellow fever became prominent areas of study [1]. Among these, a new disease termed *"dengue"* began to be investigated. In hindsight, the *"dengue"* described in the 19th century encompassed a group of arboviral diseases that were not yet understood. Today, we know that arboviral infections share common symptoms such as fever, myalgia, fatigue, and headache, while each virus also has distinct characteristics. For instance, some flaviviruses, including yellow fever virus (*Orthoflavivirus flavi*, YFV) and dengue virus (*Orthoflavivirus denguei*, DENV), are notable for their hemorrhagic manifestations [2]. DENV comprises four serotypes DENV 1 to DENV 4 that circulate in tropical regions in an endemic-epidemic pattern and can cause severe forms, including dengue hemorrhagic fever (DHF). *Alphaviruses*, on the other hand, are known for their joint tropism and include chikungunya virus (*Alphavirus chikungunya*, CHIKV), Mayaro virus (*Alphavirus mayaro*, MAYV), and O'nyong-nyong virus (*Alphavirus onyong*, ONNV). CHIKV is the most prominent representative of the *Alphavirus* genus [3]. Its transmission occurs through two main cycles:

(A)  a sylvatic cycle in Africa (and now parts of Asia), involving a variety of reservoir and vector species, resulting in sporadic cases or small outbreaks that are often asymptomatic or mildly symptomatic;

(B) a peri-urban or urban transmission cycle mainly transmitted by *Aedes aegypti* and *Aedes albopictus*, characterized by sudden, large-scale epidemics with high attack rates. These epidemics typically subside gradually.

The names, "dengue" and "dengue fever" was attached to distinct short-duration febrile diseases before their viral etiology was established. The clinical term "dengue" entered the medical literature in the American hemisphere in the 19th century describing a short-duration febrile exanthem. It was not an English word, but a Spanish homonym of the African Swahili "kidinga pepo," sometimes pronounced "denga" or "dyenga" This was the name given to epidemics of a disabling febrile arthritis, now known as chikungunya, during outbreaks in East Africa in 1823 and again in 1870 [4]. In 1881, James Christie, a Scottish-trained physician attached to the Sultan of Zanzibar, in the 1860s-70s published a report explicitly linking an 1823 East African outbreak of kidinga pepo to the outbreak in 1827 – 28 of "dengue" in the Americas [5]. In commenting on the dispersal of what we now know to be CHIKV, Christie described reports that the 1823 Zanzibar epidemic had spread East to Gujarat, India, then on to Calcutta and in 1824 to Rangoon. He added that there had been a similar spread of the 1823 "dengue" epidemic, this time westwards. In 1827, this epidemic reached St.Thomas in the West Indies. Christie states "I am of the opinion that both the disease and its designation were imported in the West Indian Islands direct from the East Coast of Africa" [5]. He noted it was then that the medical term "dengue" was introduced into the West Indies as a Spanish homynym of the African word. Another observer, contemporary to the West Indian epidemic, agreed. He wrote: "when the dengue" reached New Orleans in 1828, "the disease alluded to is supposed to have been brought from Africa, with some slaves imported into the Havana. In that place it obtained the name of Dingee, Dengue, Danga, etc. It was there very prevalent and also in Barbadoes, where it received the name of Dandy fever, from the stiffened form and dread of motion in patients" [6].

The systematic study of arboviruses began in the early 20th century, following the discovery of vector transmission and the identification of viruses [7]. The number of recognized arboviruses increased from seven in 1930 to over 350 by the 1970s, and more than 500 nowadays [8]. This research was primarily driven by the United States, first through the Rockefeller Foundation and later by Yale University in the 1960s [7]. In 1971, Dr. Carey proposed revisiting descriptions of *"dengue"* epidemics predating 1950 (date of DENV's virological identification), hypothesizing that many were actually caused by CHIKV [9]. This reinterpretation was later expanded by Dr. Halstead, who documented the 1827–1828 epidemic in the Caribbean as a likely historical chikungunya outbreak, underlining the need to revisit historical uses of the term 'dengue' and their associated clinical syndromes [10].

Given the resurgence of arbovirus epidemics over the past decade, particularly in the Americas, revisiting this historical perspective is of interest. Assuming that some of the *"dengue"* epidemics described in the 19th century were not caused by DENV, CHIKV appears as a plausible alternative explanation. It is a deliberate choice by the authors to refer specifically to focus on CHIKV rather than broader arthritogenic *Alphaviruses*. To date, no other alphavirus, including MAYV and ONNV, has caused recurrent epidemics of comparable scale and frequency to those of CHIKV.

This study aimed to reclassify the *"dengue"* epidemics of the 18th and 19th centuries using three analytical methods. To provide clarity, we use the term *"dengue"* to refer to the nosological entity described in the 19th century, based solely on symptomatic clinical observations. When we refer to DENV or CHIKV, we mean the infections as they are defined today, based on virological diagnostic methods. Although no 19th-century arbovirus epidemics were biologically confirmed, these terms are used to contextualize the findings within the framework of current scientific knowledge.

## Methods

### Bibliographical research

The study period for this research spanned 100 years, from 1779 (the year of the first scientific description of a *"dengue"* epidemic) to 1880 (the year of the last described *"dengue"* epidemic in the 19th century, excluding one recorded in Australia in 1897). A narrative review of *"dengue"* epidemics was conducted, utilizing all available articles and books related

to *"dengue"* or its synonyms. Historical encyclopedias on *"dengue"* were also consulted to supplement the bibliography [11–21].

*Inclusion criteria*:

• For the period 1831–1880: Articles that described an outbreak **and** used the term *"dengue".*"

• For the period 1779–1830 (when the term *"dengue"* was not yet in use): articles describing an epidemic that were later cited (post-1830) in encyclopedias or other sources as referring to a *"dengue"* epidemic.

*Exclusion criteria*:

• Articles not digitized.

• Absence of the author at the outbreak location.

• Insufficiently detailed clinical descriptions.

• Inconsistent clinical descriptions.

No language restrictions were applied. Digitized articles and books were sourced from various platforms, including PubMed, HathiTrust, the National Library of Scotland, Persée, the Bibliothèque numérique Medica, Smithsonian Libraries, and Google Books.

## First method: Expert opinion

Expert opinion is a common approach in historical reviews [9,10,22]. Articles were read and analyzed by a medical doctor (T.B.) with expertise in arboviruses to provide an expert perspective. This analysis aimed to propose a subjective interpretation of these *"dengue"* epidemics, considering not only the medical context but also the historical, social, and political factors of the studied period.

## Second method: Application of a clinical score

A previously developed clinical diagnostic score was used to differentiate between DENV and CHIKV infections (Fig 1). This score was based on clinical data collected during the 2014–2015 CHIKV and 2013 DENV outbreaks in French Guiana [23]. Clinical variables favouring CHIKV included joint and back pain, while those favouring DENV included headache, muscle pain, nausea/vomiting, diarrhea, and haemorrhagic signs. The score's cut-off was adjusted to achieve a sensitivity of 89% for DENV diagnosis and a specificity of 91%, improving the distinction between the two infections. DENV infection was associated with scores ranging from -9 to 0. CHIKV infection corresponded to scores ranging from +1 to +5 (Table 1). Clinical variables of interest required for score calculation were extracted from 19th-century articles.

## Third method: Double proofreading

Selected articles were randomly assigned to two adjudication committees composed of physicians specializing in infectious and tropical diseases. The first committee included infectiologists from mainland France (B.R., G.B., N.D., A.H., A.L., L.B.L.N., and A.B.), while the second consisted of physicians based in French Guiana (P.A., G.W., C.B., D.S., and L.E.) with clinical expertise in arboviral infections. The expert who had conducted the initial review (T.B.) was excluded from this process.

Each article was independently evaluated by two physicians, one from each committee, so that every article was reviewed twice by two distinct adjudicators. These physicians had unrestricted access to the full text of the articles but were blinded to each other's assessments. Based on their independent review, each adjudicator categorized the article according to one of four predefined outcomes: 1. DENV infection; 2. CHIKV infection; 3. Other pathology; 4. Insufficient information to decide.

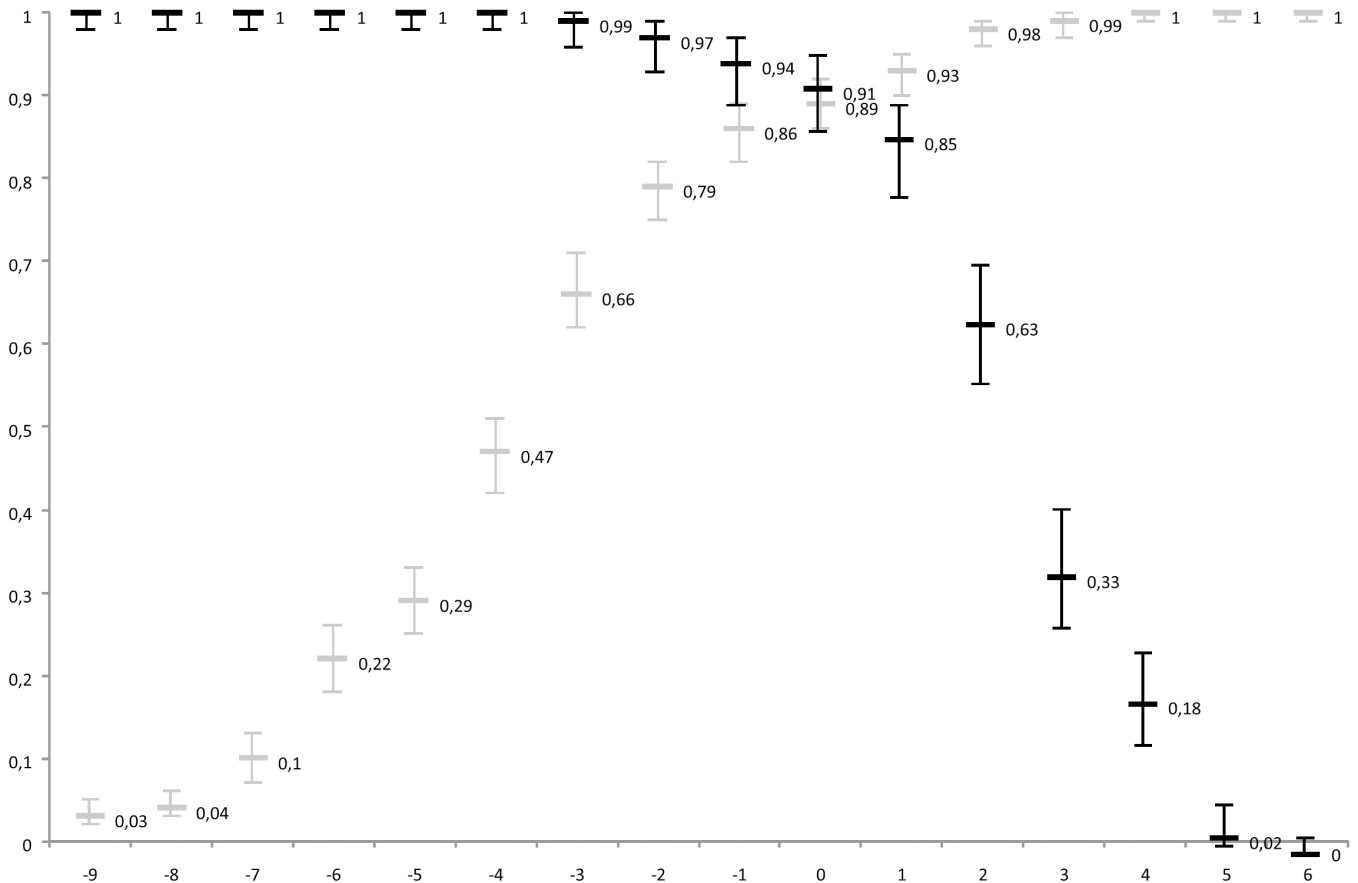

**Fig 1. Sensitivity (in grey) and specificity (in black) curves for a dengue diagnosis.** Based on values obtain from the CHIKV *vs.* DENV clinical prediction score. The figures indicate the value calculated with the 95% confidence interval.

**Table 1. Diagnostic score for DENV and CHIKV virus infection.**

| Diagnostic score DENV - CHIKV | |
|---|---|
| Headache | -1 |
| Joint pain | +5 |
| Myalgia | -2 |
| Rachialgia | +1 |
| Nausea and/or vomiting | -1 |
| Diarrhoea | -2 |
| Haemorrhage | -3 |

## Results

### Selected articles

Out of the 103 publications initially included, 41 were excluded due to the lack of a digitized version, particularly those from the 1820s describing outbreaks in India (Fig 2). Articles were published in various regional or national journals dating from the 19th century or earlier (Table 2). Of the 64 digitized articles describing epidemics, 31 were selected after applying

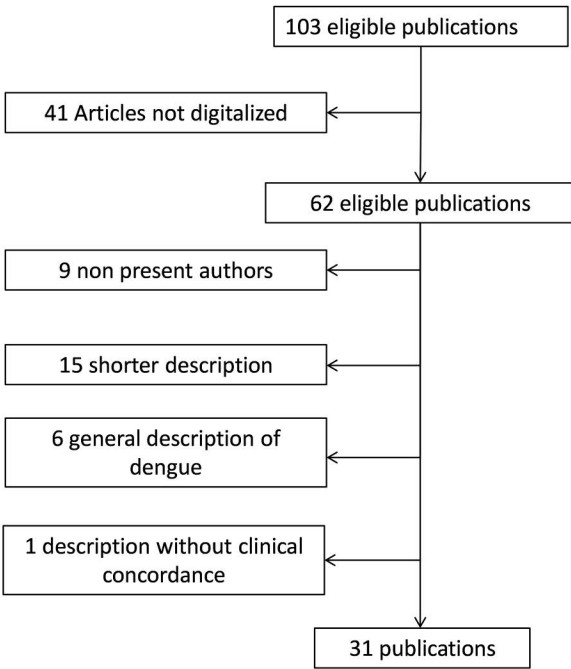

**Fig 2. Flow chart of the study.**

**Table 2. List of newspapers in which articles on "*dengue*" fever have been published.**

| List of newspapers |
| --- |
| The British Medical Journal |
| Charleston medical journal and review |
| The Edinburgh medical and surgical journal |
| The London medical and physical journal |
| The Madras monthly journal of medical science |
| New-York medical and physical journal |
| Archives de médecine navale |
| Transactions of the Medico-Chirurgical Society of Edinburgh |
| Army health reports and medical documents |
| Army health reports and medical documents |
| The Medical and surgical reporter |
| The Edinburgh medical journal |
| The Calcutta journal of medicine |
| Atlanta medical and surgical journal |
| The Southern medical and surgical journal |
| Indian medical gazette |

the exclusion criteria [4,6,16,19,24–50]. Nine of the excluded articles described outbreaks where the authors were not present at the time of the events (reported cases). Most of the selected publications focused on the epidemics of 1827–1828 and 1871–1873. The majority of the descriptions were written in English. The contributing physicians represented a

range of nationalities, including those from the United Kingdom, the United States, France, Spain, Prussia, and the Netherlands. All inter-tropical zones were represented in the analysed articles.

**First method: Expert opinion**

**General epidemiological context.** *Origin of outbreaks:* *"Dengue"* did not appear to be endemic in the Americas. At the onset of epidemics, authors often suggested an imported origin, primarily via maritime routes [32,34]. In 1827, Dr. Stedman proposed that the disease was introduced through the deep-water port of St. Thomas in the United States Virgin Islands, located in the northeastern Caribbean Sea and serving as a key gateway for ships entering the Americas. He hypothesized that *"dengue"* had been brought from the African continent. This theory was later supported by Dr. Christie, who, 50 years later, linked the disease's introduction to the transatlantic slave trade [5,33]. Similarly, the 1870 epidemic on the Indian subcontinent was believed to have been imported from Zanzibar. It is challenging to differentiate between the potential successive introductions of DENV or CHIKV. It appears that DENV was introduced in the 18th century [25]. However, it is difficult to determine whether it was circulating endemically during that period.

*Estimation of prevalence:* Several authors attempted to estimate the attack rate of *"dengue"* during outbreaks. In Havana, it was reported that 30–40% of the population contracted the disease within a six-week period [34]. During the 1871–1873 epidemic, attack rate estimates ranged from 70–80%, with some areas experiencing infection rates as high as 94% [5,47,51,52]. Most authors agreed that *"dengue"* epidemics affected individuals regardless of age, sex, or origin, suggesting a lack of herd immunity and supporting the theory that it was a non-endemic disease [31].

*Mortality:* All accounts described *"dengue"* as a highly contagious disease with an exceptionally low mortality rate [31]. During an epidemic in Cádiz, Spain, in 1784, Dr. Cubillas even referred to the disease as "la piadosa" ("the pious one") due to its mild nature [26]. This characterization stood in stark contrast to other contemporaneous epidemics, which often had significantly higher mortality rates. These epidemiological observations pointed to a mild arboviral infection (excluding YFV) but did not provide definitive evidence to differentiate between DENV and CHIKV.

**Symptoms suggestive of CHIKV or DENV infection.** *Hemorrhagic syndrome:* Due to their traumatic experiences with YFV epidemics, physicians were particularly vigilant in identifying and reporting hemorrhagic symptoms [25]. Despite tens of thousands of cases reported during the 1827–1828 and 1871–1873 periods, no hemorrhagic episodes were documented. In contrast, the 1780 and 1850 outbreaks included several cases of gingivorrhagia, metrorrhagia, hematemesis, and hemoptysis, supporting a diagnosis of DENV.

*Rash:* Rashes are frequently mentioned in the literature. Typically, the infection begins with a palmoplantar scarlet rash, sometimes accompanied by edema and/or pruritus in the affected areas. This initial rash was often followed around day 8 by a secondary measles-like rash. Both rashes could be accompanied by scaling. However, these dermatological signs were not specific enough to differentiate between the two diseases.

*Acute joint pain:* In 1779, Dr. Bylon (himself infected) described inflammatory stiffness in the ankles, with a marked increase in joint stiffness in the morning. Joint pain was a prominent feature in nearly all accounts from the 1827–1828 and 1871–1873 epidemics. Dr. Stedman, in 1827, detailed the progression of symptoms: "A person in perfect health would suddenly feel a stiffness, close to pain, in one of his fingers, most often the little finger. The stiffness increased and was followed by intense pain that spread rapidly throughout the hand, from the arm to the shoulder. In a few hours, the fingers of both hands became swollen, stiff, and painful, making any attempt to flex the joints unbearable" [33].

Joint pain emerged as the most disabling symptom of the infection, strongly favoring CHIKV. Similarly, Dr. Nicholson, in 1827, described patients as "unable to walk" with fingers so swollen that "all manual work [was] impossible" [35]. By 1873, Dr. Morice considered gouty arthritis a potential differential diagnosis [19]. In the same year, physicians in Mauritius began to regard joint swelling as a pathognomonic sign of the outbreak [49]. The presence of joint swelling, strongly associated with CHIKV, was notably absent in DENV infections.

*Chronic joint pain:* Many articles from the 1827–1828 and 1871–1873 epidemics describe chronic joint pain lasting several weeks or months, a hallmark feature of CHIKV. In contrast, during the 1780 and 1850 epidemics, the reported symptoms included diffuse pain, sometimes involving the joints, but without chronicity. These episodes presented a milder clinical picture with less emphasis on joint involvement, favoring a diagnosis of DENV.

*Atypical signs:* Some less commonly reported symptoms have also been documented. For instance, Dr. Cock noted in 1827 that "the lobes of the ears were also much affected and painful" during the initial phase of the disease [29]. This unusual manifestation may suggest ear chondritis, potentially linked to the synovial tropism characteristic of CHIKV [53].

**Historical authors interrogation.** As early as the 19th century, some physicians expressed doubts about the nature of this new disease termed *"dengue".*" Some believed the term referred to multiple distinct diseases, while others saw it as encompassing a variety of manifestations. In 1828, Dr. Ruan proposed that *"dengue"* represented a new disease, though he acknowledged its similarities to Dr. Rush's "bilious remittent fever" of 1780, now recognized as the first documented DENV epidemic [31].

Similarly, Dr. Stedman identified chronic joint pain as the key difference between *"dengue"* and bilious remittent fever (e.g., malaria), as such symptoms were absent from Dr. Rush's accounts [33]. During the 1860 *"dengue"* epidemic in Martinique, Dr. Ballot, who had previously encountered *"dengue"* in West Africa, highlighted notable distinctions: "The muscular pains and arthralgias do not seem so intense, [...] In no case were there swellings of the extremities or convalescences so long [...]" [41].

Synonyms used in various descriptions could generally be categorized into two groups: those emphasizing articular involvement and those focusing on non-articular symptoms (Tables 3 and 4). The widespread confusion reflected in many accounts supports the hypothesis that these outbreaks likely involved a mix of CHIKV and DENV cases.

## Second method: Application of a clinical score

An adapted diagnostic score was applied to the 31 selected articles, classifying 24 descriptions as CHIKV and 7 as DENV (Table 5). The epidemics classified as CHIKV included: 1779 in Indonesia, 1825 in India, 1827–1828 in the Americas, and 1871–1873 in the Indian Ocean. Three additional epidemics classified as CHIKV were isolated events: 1846 in Brazil, 1850 in the USA, and 1865 in West Africa.

In contrast, DENV epidemics were more temporally dispersed: 1780 in Philadelphia, 1784 in Spain, 1850 in the USA, 1860 in the Caribbean, and 1880 in the USA. Joint swelling, a hallmark of *Alphavirus* infection, was reported in 71% (15/21) of the CHIKV-associated descriptions but was absent from the DENV group. Notably, 70% (15/21) of CHIKV-classified descriptions had a score of 2, which lies at the threshold of differentiation. This highlights the overlapping clinical presentations of the two diseases.

DENV infections were characterized by a higher prevalence of digestive and hemorrhagic symptoms, while joint pain, although prominent, was not exclusively indicative of CHIKV. The diagnostic score results aligned with expert opinion in 97% of cases (30/31). Only one outbreak failed to match the expert classification [38], due to insufficient reporting of

**Table 3. Example of a synonym given to "*dengue*" between 1780 and 1880.**

| CHIKV | | DENV | |
|---|---|---|---|
| Ephemeral Arthritic Fever | Ore (1825) | Bilious Remitting fever | Rush (1780) |
| Rhumatismus febrilis | Squaer (1828) | Neuralgic fever | Campbel (1851) |
| Peculiar Arthritic Emanthem | Nicholson (1828) | Epidemic neuralgia | Anderson (1851) |
| Exanthesis Arthrosia | Good (1829) | Break bone fever | Arnorld (1850) |
| Joint fever in tropical countries | Thaly (1665) | | |
| Eruptive articular fever | Sheriff (1872) | | |

**Table 4. Proposed classification of "*dengue*" epidemics that occurred between 1779 and 1880 as DENV or CHIKV infection.**

| CHIKV | | | DENV | | |
|---|---|---|---|---|---|
| Bylon | Indonesia | 1779 | Rush | USA | 1780 |
| Ore | India | 1825 | Cubillas | Spain | 1784 |
| Cock | West Indies | 1827 | De castilla | Spain | 1784 |
| Maxwell | West Indies | 1827 | Anderson | USA | 1850 |
| Ruan | West Indies | 1827 | Arnold | USA | 1850 |
| Squaer | West Indies | 1827 | Campbell | USA | 1850 |
| Stedman | West Indies | 1827 | Ballot | West Indies | 1860 |
| Dickson | USA | 1828 | Horlbeck | USA | 1880 |
| Dumaresq | USA | 1828 | | | |
| Lehman | West Indies | 1828 | | | |
| Nicholson | West Indies | 1828 | | | |
| Bennett | USA | 1829 | | | |
| Lallement | Brazil | 1846 | | | |
| Thaly | West Africa | 1865 | | | |
| Christie | East Africa | 1871 | | | |
| Vauvray | Egypt | 1871 | | | |
| Dunkley | India | 1872 | | | |
| Sheriff | India | 1872 | | | |
| Sparrow | India | 1872 | | | |
| Wise | India | 1872 | | | |
| Cotholendy | Reunion island | 1873 | | | |
| Labonté | Mauritius | 1873 | | | |
| Morice | Vietnam | 1873 | | | |

DENV-specific symptoms, particularly hemorrhagic or digestive signs. Considering the context and the geographic proximity of the outbreak to other affected areas (Georgia and South Carolina), it was likely a DENV outbreak.

### Third method: Double Proofreading by specialists in infectious and tropical diseases from French Guiana and mainland France

A review of the 31 articles was conducted by two groups of infectious and tropical disease specialists: one based in French Guiana (5 physicians) and the other in mainland France (7 physicians) (Table 6). According to this method, 61% (n = 19/31) of the diagnoses were concordant between the two groups, while 26% (n = 8/31) were discordant. In 13% of cases (n = 4/31), one of the reviewers indicated insufficient clinical evidence to reach a conclusion.

Out of the 31 epidemic descriptions analyzed, 20 (65%) showed full agreement across all three methods (expert opinion, score, and both reviewer groups), consistently pointing to either CHIKV or DENV. These cases support the robustness of the combined approach. In contrast, 11 descriptions presented varying degrees of disagreement. In 8 of them, the reviewers from French Guiana and mainland France diverged in their classification; in 3 additional cases, there was disagreement between one or both reviewer groups and the clinical score. A comparison of the classifications made by the two groups of reviewers—those based in mainland France and those practicing in French Guiana—revealed a higher level of agreement with the expert opinion among the French Guiana group (83%) than among the mainland France group (75%) (Fig 3).

The underlying causes of discordance appear multifactorial, including limited clinical detail, ambiguities in symptom descriptions, or overlapping features such as the presence of fever and arthralgia without chronicity. These discrepancies

**Table 5. Application of the DENV-CHIKV diagnostic score to the 31 "*dengue*" epidemic descriptions included in the study.**

| Authors | Region | Year | Total | Result | Arthritis |
|---|---|---|---|---|---|
| Bylon | Indonesia | 1779 | 2 | CHIKV | 1 |
| Rush | USA | 1780 | -1 | DENV | 0 |
| Cubillas | Spain | 1784 | 0 | DENV | 0 |
| De Castilla | Spain | 1784 | -4 | DENV | 0 |
| Ore | India | 1825 | 2 | CHIKV | 1 |
| Cock | West Indies | 1827 | 5 | CHIKV | 0 |
| Maxwell | West Indies | 1827 | 2 | CHIKV | 1 |
| Ruan | West Indies | 1827 | 2 | CHIKV | 0 |
| Squaer | West Indies | 1827 | 2 | CHIKV | 1 |
| Stedman | West Indies | 1827 | 2 | CHIKV | 1 |
| Dickson | USA | 1828 | 2 | CHIKV | 1 |
| Dumaresq | USA | 1828 | 2 | CHIKV | 1 |
| Lehman | West Indies | 1828 | 3 | CHIKV | 0 |
| Nicholson | West Indies | 1828 | 2 | CHIKV | 1 |
| Bennett | USA | 1829 | 3 | CHIKV | 0 |
| Lallement | Brazil | 1846 | 3 | CHIKV | 1 |
| Anderson | USA | 1850 | 2 | CHIKV | 0 |
| Arnold | USA | 1850 | -3 | DENV | 0 |
| Campbell | USA | 1850 | -9 | DENV | 0 |
| Ballot | West Indies | 1860 | -2 | DENV | 0 |
| Thaly | West Africa | 1865 | 5 | CHIKV | 0 |
| Christie | East Africa | 1871 | 4 | CHIKV | 1 |
| Vauvray | Egypt | 1871 | 2 | CHIKV | 1 |
| Dunkley | India | 1872 | 2 | CHIKV | 1 |
| Sheriff | India | 1872 | 3 | CHIKV | 1 |
| Sparrow | India | 1872 | 2 | CHIKV | 1 |
| Wise | India | 1872 | 3 | CHIKV | 0 |
| Cotholendy | Reunion island | 1873 | 2 | CHIKV | 0 |
| Labonté | Mauritius | 1873 | 4 | CHIKV | 1 |
| Morice | Vietnam | 1873 | 2 | CHIKV | 1 |
| Horlbeck | USA | 1880 | 0 | DENV | 0 |

highlight the intrinsic difficulty of diagnosing historical syndromes retrospectively without virological confirmation and support the need for a triangulated approach that cross-validates finding. Ultimately, this layered strategy provides a more nuanced interpretation of 19th-century epidemic data and illustrates the value of cross-methodological comparison in historical epidemiology.

## Discussion

This study sought to retrospectively reclassify 18th and 19th century "*dengue*" epidemics using three diagnostic approaches. While these methods cannot replace biological confirmation—which is inherently impossible in this context—they each provide valuable insight. Concerning the first method, expert opinion is currently the most commonly used approach in historical studies of medicine. Despite its subjective nature, this approach allows for a comprehensive and contextual interpretation of historical medical texts. It enables the integration of linguistic nuances, social context, and clinical reasoning in the absence of standardized definitions or laboratory confirmation. Its continued use in the field reflects

**Table 6. Diagnoses retained after reading the 31 articles describing "*dengue*" epidemics in the 19th century.**

| Authors | Region | Year | Diagnosis by score | Mainland readers | Guiana readers | Discordance | Diagnosis by expert opinion |
|---------|--------|------|--------------------|-----------------|----------------|-------------|----------------------------|
| Anderson | USA | 1850 | CHIKV | DENV | DENV | Yes | DENV |
| Thaly | West Africa | 1865 | CHIKV | DENV | DENV | No | CHIKV |
| Morice | Vietnam | 1873 | CHIKV | DENV | DENV | Yes | CHIKV |
| Ore | India | 1825 | CHIKV | CHIKV | DENV | Yes | CHIKV |
| Dumaresq | USA | 1828 | CHIKV | DENV | CHIKV | Yes | CHIKV |
| Lehman | West Indies | 1828 | CHIKV | CHIKV | DENV | Yes | CHIKV |
| Bennett | USA | 1829 | CHIKV | DENV | CHIKV | Yes | CHIKV |
| Vauvray | Egypt | 1871 | CHIKV | DENV | CHIKV | Yes | CHIKV |
| Sheriff | India | 1872 | CHIKV | DENV | CHIKV | Yes | CHIKV |
| Cotholendy | Reunion Island | 1873 | CHIKV | DENV | CHIKV | Yes | CHIKV |
| Horlbeck | USA | 1880 | DENV | DENV | CHIKV | Yes | DENV |
| De Castilla | Spain | 1784 | DENV | NA | DENV | Yes | DENV |
| Dickson | USA | 1828 | CHIKV | NA | CHIKV | Yes | CHIKV |
| Ballot | West Indies | 1860 | DENV | NA | DENV | Yes | DENV |
| Christie | East Africa | 1871 | CHIKV | CHIKV | NA | Yes | CHIKV |
| Bylon | Indonesia | 1779 | CHIKV | CHIKV | CHIKV | No | CHIKV |
| Rush | USA | 1780 | DENV | DENV | DENV | No | DENV |
| Cubillas | Spain | 1784 | DENV | DENV | DENV | No | DENV |
| Cock | West Indies | 1827 | CHIKV | CHIKV | CHIKV | No | CHIKV |
| Maxwell | West Indies | 1827 | CHIKV | CHIKV | CHIKV | No | CHIKV |
| Ruan | West Indies | 1827 | CHIKV | CHIKV | CHIKV | No | CHIKV |
| Squaer | West Indies | 1827 | CHIKV | CHIKV | CHIKV | No | CHIKV |
| Stedman | West Indies | 1827 | CHIKV | CHIKV | CHIKV | No | CHIKV |
| Nicholson | West Indies | 1828 | CHIKV | CHIKV | CHIKV | No | CHIKV |
| Lallement | Brazil | 1846 | CHIKV | CHIKV | CHIKV | No | CHIKV |
| Arnold | USA | 1850 | DENV | DENV | DENV | No | DENV |
| Campbell | USA | 1850 | DENV | DENV | DENV | No | DENV |
| Dunkley | India | 1872 | CHIKV | CHIKV | CHIKV | No | CHIKV |
| Sparrow | India | 1872 | CHIKV | CHIKV | CHIKV | No | CHIKV |
| Wise | India | 1872 | CHIKV | CHIKV | CHIKV | No | CHIKV |
| Labonté | Mauritius | 1873 | CHIKV | CHIKV | CHIKV | No | CHIKV |

Mainland readers correspond to readers practising in mainland France and Guiana readers to readers practising in French Guiana. In grey were figured situation when a reader was not able (NA) to propose a diagnosis, in red discordance between the 2 responses and orange discordance between reader results and diagnosis by score. DENV: Dengue virus, CHIKV: chikungunya virus.

its relevance for reconstructing past medical events when objective data are lacking. This method is scientifically limited in robustness but allows for the development of plausible hypotheses. Notably, Dr. Carey and, later, researchers such as Dr. Halstead used this approach to challenge long-standing misconceptions from the 19th century — including the belief that there were no historical CHIKV epidemics, and the misuse of the term "*dengue*" prior to the advent of serological testing.

The clinical score allowed a standardized comparison based on reported symptoms, though it required detailed descriptions. It is important to note that the limitations of the diagnostic score are similar to those encountered in its contemporary use. The common clinical presentations of arboviral infections remain relatively comparable, and differentiation mainly relies on the presence of hemorrhagic signs for DENV—signs which are generally considered reliable in historical reports, as they were alarming and thus often described in detail. The presence of arthralgia in favor of CHIKV may be

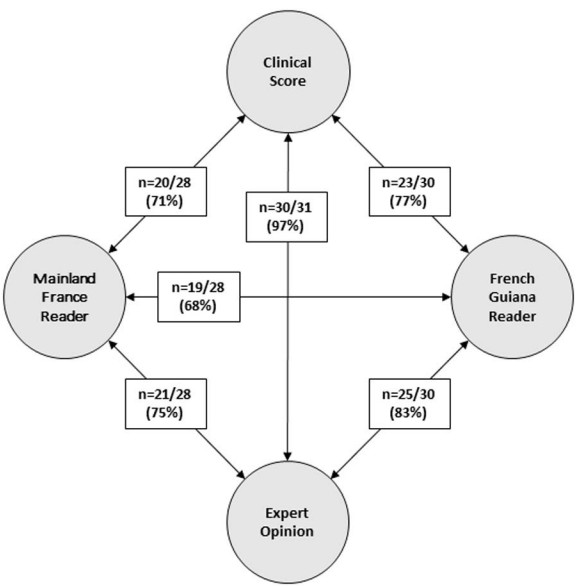

**Fig 3. Concordance between diagnostic methods.** Figure legend: Cases with missing data (i.e., where one of the reviewers did not provide a diagnosis) were excluded from the analysis.

more debatable, especially in the early 19th century. However, most authors explicitly mention chronic joint pain. Rheumatism was well known and described at the time.

The dual review by physicians from two geographic and epidemiological backgrounds increased objectivity, but revealed substantial heterogeneity in interpretation. This difference may reflect the greater familiarity of French Guiana physicians with arboviral infections, which are endemic in their region. In contrast, reviewers from mainland France may be less frequently exposed to these infections in routine practice. It is also important to note that the expert opinion was provided by a physician practicing in French Guiana, and the diagnostic score was developed by authors also based in this region — which may represent a source of bias.

Regarding concordance, as noted, there appeared to be differences potentially linked to prior experience with arboviral infections. However, few consistent explanatory factors were identified. The presence of arthritis, for example, was not systematically interpreted as suggestive of CHIKV by reviewers. Dr. Morice's account, which compared the "*dengue*" epidemic to gout attacks, was nonetheless classified as a probable DENV epidemic by both reviewers. Conversely, the 1850 epidemic described by Dr. Anderson was classified as CHIKV due to the presence of joint pain, even though no hemorrhagic signs were reported. Most full-agreement classifications were made in the context of detailed and comprehensive clinical descriptions. Unfortunately, regardless of the method used, each reader is inevitably influenced by the clinical lens of the original author. This is precisely why second-hand reports were excluded from the study—to minimize this inherent bias.

In summary, the analysis suggests that the epidemics of 1827–29 in the Americas and 1871–73 in the Indian Ocean may have been caused by chikungunya virus (CHIKV) rather than dengue virus (DENV). The consistent reports of disabling joint symptoms and swelling across multiple sources support this hypothesis. In contrast, the 1780 and 1850 outbreaks in the United States presented a more classical dengue-like syndrome, with hemorrhagic and digestive signs more typical of DENV infection. Some unexplained cases, such as the 1846 epidemic in Brazil, raise the question of whether other alphaviruses—such as Mayaro virus (MAYV), which is endemic in South America—might have been responsible, although MAYV has not been known to cause large-scale epidemics.

This retrospective approach carries important limitations. In particular, one must avoid the epistemological bias of presentism—the tendency to interpret historical phenomena through the lens of contemporary concepts and concerns. Although we applied modern nosological categories and Western medical standards to 19th-century descriptions, this period remains close to our present time, and modern biomedicine emerged from it [54]. Moreover, despite likely genetic variability, both CHIKV and DENV in the 19th century may have been clinically similar to the viruses we know today. This is supported by over 75 years of clinical observation showing remarkable consistency in symptomatology and transmission dynamics. While severe or atypical forms are better recognized today, the core presentation has not changed. Even now, in epidemic settings, clinical diagnosis remains widely used when molecular tools are limited.

The apparent underrepresentation of DENV outbreaks in our findings could be due to historical reporting bias. DENV likely circulated endemically in Asia and Africa during the 19th century, but Western physicians—mostly located in coastal trading posts—may have underreported its activity, particularly outside of large urban outbreaks. Endemic DENV may also have been confused with other febrile illnesses such as yellow fever or malaria. Conversely, CHIKV's distinctive clinical profile and the size of its epidemics may have made it more visible in historical records. While other arboviruses such as O'nyong-nyong virus (ONNV) or Oropouche virus (OROV) have epidemic potential, only CHIKV and DENV have demonstrated repeated large-scale international spread. The recurring similarity in epidemic dynamics over the last two centuries suggests that lessons from the past may still be relevant for understanding present and future emerging diseases.

These methods could also be applied to later epidemics between 1880 and 1950, the etiology of which remains debated, as in the case of the 1900 epidemic in Singapore, which spread throughout South Asia and is still often attributed to dengue fever. However, its regional spread and limited clinical descriptions suggest that it may have been caused by CHIKV rather than DENV.

## Conclusion

Recognizing the interpretative challenges in revisiting ancient medical texts, this study employed three distinct diagnostic methods. First, expert opinion provided a contextualized and comprehensive analysis but was subject to criticism due to its subjective nature. Second, the application of a clinical score allowed for greater objectivity and a more stringent diagnosis, although it was limited by the need for a complete description. Finally, proofreading by specialists contributed to greater objectivity but resulted in greater heterogeneity of results. When applied to descriptions of 19th-century *"dengue"* epidemics, all three methods were consistent in suggesting that a significant proportion of these epidemics were likely caused by CHIKV rather than DENV. The DENV epidemics were primarily those of 1780 in Philadelphia and 1850 in the southern United States, while the possible CHIKV epidemics were of a much larger scale, affecting the entire continental Americas in 1827–29, reminiscent of the 2013–15 epidemic, and the Indian Ocean and Indian subcontinent in 1871–73, reminiscent of the 2004–2006 epidemics.

Without modern serological or molecular tests, it was impossible to definitively diagnose these arboviruses. This raises the question of whether epidemics with "arthritic" symptoms were caused by CHIKV or an indigenous alphavirus like MAYV, which has a similar clinical profile but does not cause large-scale outbreaks like CHIKV. Current patterns point to CHIKV rather than MAYV. Studies have also shown that CHIKV epidemics in the 19th century led to the emergence of new strains [55]. While the 19th-century epidemics could be linked to CHIKV, certainty is impossible. This historical review is important, especially as researchers today search for the next emerging virus that could cause a pandemic.

## Acknowledgments

Thanks to the teams from the Infectious and Tropical Diseases Department and the Clinical Investigation Center at Cayenne Hospital for their support in research on arboviruses. Thanks to Dr. Rousset from the Institut Pasteur of French Guiana.

## Author contributions

**Conceptualization:** Timothee Bonifay, Loïc Epelboin.

**Investigation:** Clémence Bonnefoy, Benjamin Rossi, Edouard Hallet, Philippe Abboud, Guillaume Bellaud, Nathalie Dournon, Aurélia Henn, Adrien Lemaignen, Liem Binh Luong Nguyen, Diane Sanderink, Gaelle Walter, Alexandre Bleibtreu.

**Methodology:** Timothee Bonifay, Edouard Hallet.

**Resources:** Clémence Bonnefoy.

**Supervision:** Loïc Epelboin.

**Writing – original draft:** Timothee Bonifay, Clémence Bonnefoy, Loïc Epelboin.

**Writing – review & editing:** Mathieu Nacher.

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
