## [Decision Letter · Decision Letter 0]

29 Apr 2025

Comparing Diagnostic Methods for Historical Arbovirus Outbreaks: Insights from 19th Century “dengue” Epidemics

Dear Dr. Bonifay,

Thank you for submitting your manuscript to PLOS Neglected Tropical Diseases. After careful consideration, we feel that it has merit but does not fully meet PLOS Neglected Tropical Diseases's publication criteria as it currently stands. Therefore, we invite you to submit a revised version of the manuscript that addresses the points raised during the review process.

Please submit your revised manuscript within 60 days Jun 28 2025 11:59PM. If you will need more time than this to complete your revisions, please reply to this message or contact the journal office at plosntds@plos.org. Please include the following items when submitting your revised manuscript:

We look forward to receiving your revised manuscript.

Kind regards,

Joshua Anzinger

Academic Editor

David Safronetz

Section Editor

Shaden Kamhawi

co-Editor-in-Chief

Paul Brindley

co-Editor-in-Chief

**Additional Editor Comments:**

The authors should ensure that all reviewer comments are addressed and that formatting of the manuscript is in accordance with PLOS Neglected Tropical Diseases guidelines.

**Journal Requirements:**

At this stage, the following Authors/Authors require contributions: Timothee Bonifay, Mathieu Nacher, Clémence Bonnefoy, Benjamin Rossi, Edouard Hallet, Philippe Abboud, Guillaume Bellaud, Nathalie Dournon, Aurélia Henn, Adrien Lemaignen, Liem Binh Luong Nguyen, Diane Sanderink, Gaelle Walter, Alexandre Bleibtreu, and Loïc Epelboin. Please ensure that the full contributions of each author are acknowledged in the "Add/Edit/Remove Authors" section of our submission form.

4) We note that your Data Availability Statement is currently as follows: "All data used in the study are available online.". Please confirm at this time whether or not your submission contains all raw data required to replicate the results of your study. Authors must share the “minimal data set” for their submission. PLOS defines the minimal data set to consist of the data required to replicate all study findings reported in the article, as well as related metadata and methods (https://journals.plos.org/plosone/s/data-availability#loc-minimal-data-set-definition).

- The points extracted from images for analysis..

**Reviewers' Comments:**

Reviewer's Responses to Questions

**Key Review Criteria Required for Acceptance?**

**Methods**

-Are the objectives of the study clearly articulated with a clear testable hypothesis stated?

-Is the study design appropriate to address the stated objectives?

-Is the population clearly described and appropriate for the hypothesis being tested?

-Is the sample size sufficient to ensure adequate power to address the hypothesis being tested?

-Were correct statistical analysis used to support conclusions?

-Are there concerns about ethical or regulatory requirements being met?

Reviewer #1: (No Response)

Reviewer #2: (No Response)

Reviewer #3: yes. But, the authors historical debt to other writers has not been acknowledged.

**Results**

-Does the analysis presented match the analysis plan?

-Are the results clearly and completely presented?

-Are the figures (Tables, Images) of sufficient quality for clarity?

Reviewer #1: (No Response)

Reviewer #2: (No Response)

Reviewer #3: yes

**Conclusions**

-Are the conclusions supported by the data presented?

-Are the limitations of analysis clearly described?

-Do the authors discuss how these data can be helpful to advance our understanding of the topic under study?

-Is public health relevance addressed?

Reviewer #1: (No Response)

Reviewer #2: (No Response)

Reviewer #3: yes

**Editorial and Data Presentation Modifications?**

Reviewer #1: (No Response)

Reviewer #2: (No Response)

Reviewer #3: comments and suggestions are made in other locations.

**Summary and General Comments**

Reviewer #1: This manuscript by Bonifay and colleagues presenting an interesting historical analysis of past outbreaks of "dengue", for which the authors investigate evidence that they were actually caused by dengue virus, chikungunya virus, or possible other pathogens. This is a topic that frequently comes into discussion in the field, and hence the authors systematic approach to the question is a useful contribution to the field. Generally, they acknowledge and accept the various limitations of such an endeavor and do not overstate their findings or conclusions. Nonetheless, several revisions are needed:

1) Section 2.4: the authors state that "Each article was independently reviewed by one physician from each [adjudication] group, ensuring two readings per article. After reviewing the articles, the adjudicators categorized them". It is not clear from this if each article was categorized by the two physicians reviewing the article, or if the physicians presented their interpretation to the adjudication committee who then together determined the categorization, or something else. Additional information is needed to clarify this process. Of note, if only one physician classified each article for each committee, it calls into question the validity and utility of this part of the overall study.

2) The calculation of sensitivity, specificity, and accuracy should be removed from the manuscript, as there is no "gold standard" by which to compare any of the analytic components of this study. One could make a case for any of the three parts (scoring, readings, proofreading) to be compared to any other - I see no reason why the expert opinion proofreading should be seen as the most correct and used as the comparator. That said, I would appreciate if the authors were to present an analysis of which outbreaks were determined to be most likely caused by CHIKV or DENV via agreement of all three methods vs. those for which there were disagreement. (The authors present some such comparison already, but not agreement between all methods.)

3) The format of the manuscript as a whole is atypical and (subject to the direction of the journal editors) may require significant re-formatting. Of note:

a) The use of numbered section headings (1, 2.3, 3.2.4, etc.) is not typical. Standard headings (Introduction, Materials and Methods, etc.) followed by non-numbered headings may be of greater utility to the reader.

b) The abstract does not follow the standard format of Intro/Methods/Results/Conclusions. I defer to the editors to determine if re-formatting is required or if the current format is acceptable.

c) Lines 230-232 belong in the Discussion.

d) Lines 251-255 belong in the Discussion.

e) Section 4 (Limitations) should be a component of the Discussion, not a free-standing section.

Reviewer #2: In this article, Bonifay et al attempt to assign the probably causal infectious etiology of multiple historic “dengue” outbreaks. Using a series of literature reviews, scoring metrics, and input from current infectious disease clinicians, the authors posit that many historic “dengue” outbreaks that occurred before the identification of DENV may have been caused by other arboviruses, such as CHIKV.

My most significant concern with the methodology utilized in this study is that the clinical presentation of arboviral infections – such as DENV and CHIKV – can vary significantly over time and in different regions of the world, and additionally based on the exposure history of the population being examined. I have no doubt that the experts polled in this study are extremely competent in identifying the clinical symptoms of modern DENV and CHIKV infections, but it seems like a significant leap to assume that the clinical presentation of these infections would be identical 200+ years ago given wide variation in clinical presentation we see today. We know from modern serostudies that clinical diagnosis of even modern DENV infections can be very hit-or-miss without molecular, serologic, and/or antigenic diagnostics. It seems hard to believe that 2nd/3rd-hand diagnosis would be any more accurate.

Reviewer #3: The authors have compiled a reasonably complete bibliography of the nearly 250-year-old literature describing outbreaks of short duration febrile diseases caused by chikungunya and dengue viruses. It is not merely confusing, but somewhat short-shrifting for the authors to fail to preface this review by acknowledging dengue-expert Donald Carey’s 1970 trip to the Harvard Library where he had recognized that the 1779 “dengue” epidemic in Batavia, New Netherlands closely resembled chikungunya febrile disease. In the medical literature, it had long been labeled an outbreak of dengue. From this insight, he wrote “Chikungunya and “dengue,” a case of mistaken identity?” published in 1971. Equally important, a report by another arbovirologist, Scott Halstead, published in 2015, recognized that chikungunya in the Caribbean was a reappearance of a chikungunya virus that had been transported from East Africa in 1827 to the Caribbean where the disease it caused had been given the name, “dengue.” This article, rich in historical details, was published in 2015.

PLOS authors have the option to publish the peer review history of their article (what does this mean? ). If published, this will include your full peer review and any attached files.

**Do you want your identity to be public for this peer review?** For information about this choice, including consent withdrawal, please see our Privacy Policy .

Reviewer #1: No

Reviewer #2: No

Reviewer #3: **Yes: ** Scott B Halstead

**Figure resubmission:**

**Reproducibility:**



---

## [Decision Letter · Decision Letter 1]

11 Aug 2025

Comparing Diagnostic Methods for Historical Arbovirus Outbreaks: Insights from 19th Century “dengue” Epidemics

Dear Dr. Bonifay,

Thank you for submitting your manuscript to PLOS Neglected Tropical Diseases. After careful consideration, we feel that it has merit but does not fully meet PLOS Neglected Tropical Diseases's publication criteria as it currently stands. Therefore, we invite you to submit a revised version of the manuscript that addresses the points raised during the review process.

Please submit your revised manuscript within 30 days Sep 10 2025 11:59PM. If you will need more time than this to complete your revisions, please reply to this message or contact the journal office at plosntds@plos.org. Please include the following items when submitting your revised manuscript:

* A rebuttal letter that responds to each point raised by the editor and reviewer(s). You should upload this letter as a separate file labeled 'Response to Reviewers '. This file does not need to include responses to any formatting updates and technical items listed in the 'Journal Requirements' section below.

* A marked-up copy of your manuscript that highlights changes made to the original version. You should upload this as a separate file labeled 'Revised Manuscript with Track Changes '.

* An unmarked version of your revised paper without tracked changes. You should upload this as a separate file labeled 'Manuscript '.

We look forward to receiving your revised manuscript.

Kind regards,

Joshua Anzinger

Academic Editor

David Safronetz

Section Editor

Shaden Kamhawi

co-Editor-in-Chief

Paul Brindley

co-Editor-in-Chief

**Additional Editor Comments:**

Please address Reviewer 3’s comments completely.

**Journal Requirements:**

**Reviewers' comments:**

Reviewer's Responses to Questions

**Key Review Criteria Required for Acceptance?**

**Methods**

-Are the objectives of the study clearly articulated with a clear testable hypothesis stated?

-Is the study design appropriate to address the stated objectives?

-Is the population clearly described and appropriate for the hypothesis being tested?

-Is the sample size sufficient to ensure adequate power to address the hypothesis being tested?

-Were correct statistical analysis used to support conclusions?

-Are there concerns about ethical or regulatory requirements being met?

Reviewer #1: (No Response)

Reviewer #3: Okay

**Results**

-Does the analysis presented match the analysis plan?

-Are the results clearly and completely presented?

-Are the figures (Tables, Images) of sufficient quality for clarity?

Reviewer #1: (No Response)

Reviewer #3: yes

**Conclusions**

-Are the conclusions supported by the data presented?

-Are the limitations of analysis clearly described?

-Do the authors discuss how these data can be helpful to advance our understanding of the topic under study?

-Is public health relevance addressed?

Reviewer #1: (No Response)

Reviewer #3: yes

**Editorial and Data Presentation Modifications?**

Reviewer #1: (No Response)

Reviewer #3: The reviewer offers as a “gift” sentences from his own history of chikungunya. “The names, “Dengue” and “dengue fever” were attached to distinct short-duration febrile diseases before their viral etiology was established. The clinical term “dengue” entered the medical literature in the American hemisphere in the 19th century describing a short-duration febrile exanthem. It was not an English word, but a Spanish homonym of the African Swahili “kidinga pepo,” sometimes pronounced “denga” or “dyenga” This was the name given to epidemics of a disabling febrile arthritis, now known as chikungunya, during outbreaks in East Africa in 1823 and again in 1870. (1) In 1881, James Christie, a Scottish-trained physician attached to the Sultan of Zanzibar, in the 1860s-70s published a report explicitly linking an 1823 East African outbreak of kidinga pepo to the outbreak in 1827 – 28 of “dengue” in the Americas. (2) In commenting on the dispersal of what we now know to be chikungunya virus, Christie described reports that the 1823 Zanzibar epidemic had spread East to Gujarat, India, then on to Calcutta and in 1824 to Rangoon. He added that there had been a similar spread of the 1823 “dengue” epidemic, this time westwards. In 1827, this epidemic reached St.Thomas in the West Indies. Christie states “I am of the opinion that both the disease and its designation were imported in the West Indian Islands direct from the East Coast of Africa.” (2) He noted it was then that the medical term “dengue” was introduced into the West Indies as a Spanish homynym of the African word. Another observer, contemporary to the West Indian epidemic, agreed. He wrote: “when the dengue” reached New Orleans in 1828, “the disease alluded to is supposed to have been brought from Africa, with some slaves imported into the Havana. In that place it obtained the name of Dingee, Dengue, Danga, etc. It was there very prevalent and also in Barbadoes, where it received the name of Dandy fever, from the stiffened form and dread of motion in patients.” (3)

1. Christie J. Remarks on "kidinga Pepo"' a peculiar form of exantematous disease. Epidemic in Zanzibar, East Coast of Africa, from July 1870 till January 1871. BMJ. 1872;1:577-79.

2. Christie J. On epidemics of dengue fever: their diffusion and etiology. Glasgow Med J. 1881;3:161-76.

3. Dumaresq PJ. An account of dengue, danga or dandy fever, as it occurred in New Orleans and in the person of the writer,communicated in a letter to one of the editors. . Boston Medical and Surgical Journal. 1828;1:497-502.

The reviewer hopes the authors will take the time to investigate the outbreak of 1900 in Singapore which extended throughout southern Asia. This has been confused over and over as caused by a dengue virus. Clearly, it is chikungunya. Remember, chikungunya is an African zoonosis that spills over into an urban cycle on the East coast of Africa at approximately 40 year intervals. It has made the transit across the Indian Ocean many, many times.

**Summary and General Comments**

Reviewer #1: My comments have been adequately addressed. I still have minor concerns about the format of the manuscript, but those can be addressed by the editorial staff during subsequent steps.

Reviewer #3: As suggested, this history could be expanded, helping the dengue community better understand its past.

PLOS authors have the option to publish the peer review history of their article (what does this mean? ). If published, this will include your full peer review and any attached files.

**Do you want your identity to be public for this peer review?** For information about this choice, including consent withdrawal, please see our Privacy Policy .

Reviewer #1: No

Reviewer #3: No

**Figure resubmission:**
---

## [Editor Report · Decision Letter 2]

9 Sep 2025

Dear Dr. Bonifay,

We are pleased to inform you that your manuscript 'Comparing Diagnostic Methods for Historical Arbovirus Outbreaks: Insights from 19th Century “dengue” Epidemics' has been provisionally accepted for publication in PLOS Neglected Tropical Diseases.

Best regards,

Joshua Anzinger

Academic Editor

David Safronetz

Section Editor

Shaden Kamhawi

co-Editor-in-Chief

Paul Brindley

co-Editor-in-Chief

---

## [Editor Report · Acceptance letter]

Dear Dr. Bonifay,

We are delighted to inform you that your manuscript, "Comparing Diagnostic Methods for Historical Arbovirus Outbreaks: Insights from 19th Century “dengue” Epidemics," has been formally accepted for publication in PLOS Neglected Tropical Diseases.

Best regards,

Shaden Kamhawi

co-Editor-in-Chief

Paul Brindley

co-Editor-in-Chief
